# Mineralogical and Geochemical Characteristics of Triassic Lithium-Rich K-Bentonite Deposits in Xiejiacao Section, South China

**Yongjie Lin [1], Mianping Zheng [1,*], Yongsheng Zhang [1], Enyuan Xing [1], Simon A. T. Redfern [2,3]** 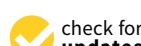 **, Jianming Xu [1], Jiaai Zhong [1] and Xinsheng Niu [1]**

[1] MNR Key Laboratory of Saline Lake Resources and Environments, Institute of Mineral Resources, Chinese Academy of Geological Sciences, Beijing 100037, China; linyj@cags.ac.cn (Y.L.); zys_601@126.com (Y.Z.); xingenyuan@cags.ac.cn (E.X.); xujianming1965@sina.com (J.X.); zhongja555@163.com (J.Z.); xs_niu@126.com (X.N.)

[2] Asian School of the Environment, Nanyang Technological University, Singapore 639798, Singapore; Simon.Redfern@ntu.edu.sg

[3] Department of Earth Sciences, University of Cambridge, Downing Street, Cambridge CB2 3EQ, UK

\* Correspondence: Zhengmp2010@126.com

**Abstract:** Widespread alteration in the Early–Middle Triassic volcanic ash of the Xiejiacao section, south China, has resulted in significant occurrences of lithium-rich K-bentonite deposits with economic potential. Detailed mineralogical and geochemical investigations of Li-rich K-bentonite deposits from the Xiejiacao section of Guangan city, South China, are presented here. The X-ray diffraction (XRD) data and major element chemistry indicates that the Li-rich K-bentonite deposits contain quartz, clay minerals, feldspar, calcite and dolomite, and the clay minerals are dominated by illite and ordered (R3) illite/smectite (I/S). The concentrations of major and trace elements in Li-rich K-bentonite deposits altered from volcanic ashes are most likely derived from felsic magmas, associated with intense volcanic arc activity. The composition of the clay components suggests that the Li-rich K-bentonite deposits are probably altered from the smectite during diagenesis, whereas smectite is mainly formed by submarine alterations of volcanic materials and subsequently the I/S derived from the volcanogenic smectite illitization. Moreover, accurate determination of the structure in I/S reveals that the temperatures reached by the sedimentary series are around 180 °C with a burial depth of ~6000 m. The widely distributed lithium-rich clay deposits strongly indicate widespread eruptions of volcanic ashes in the Early–Middle Triassic, which released huge amounts of volcanic ash. Lithium fixed in the illite and I/S is considered to have leached from the volcanogenic products by a mixed fluid source (i.e., meteoric, porewater and hydrothermal fluids). These Li-rich clay minerals in the marine basin contain economically extractable levels of metal and are a promising new target for lithium exploration.

**Keywords:** Li-rich K-bentonite deposits; geochemistry; mineralogy; Early–Middle Triassic; South China

## 1. Introduction

The increasing demand for lithium in rechargeable batteries, especially for electric vehicles, has attracted a great deal of interest in the search for more potential lithium resources. More recently, Li-rich clays have been now recognized as a significant lithium resource, following the recent assessment of the Li clay deposit in McDermitt/Kings Valley of Nevada, United States and Sonora, Mexico [1]. Although extensive studies have been conducted worldwide on clay deposits of volcanic origin



and weathering origin, the global occurrence of lithium-rich clay deposits has rarely been reported, especially in large marine basins [2,3]. "Mung bean rock" (or "green bean rock"), a type of Li-rich clay deposit, is distributed widely in Early–Middle Triassic strata of south China, over an area of around 700,000 km$^2$,and dozens of centimeters to tens of meters in thickness [4]. It is so called "mung bean rock" because of its green color and often contains siliceous clasts [5,6]. Several Li-rich clay beds have been found in the outcrop of Early–Middle Triassic strata in Sichuan Basin [7], however, only limited detailed studies have been published about the mineralogical and geochemical characteristics of mung bean rock. In particular little is known about their formation conditions and the supernormal enrichment mechanism of lithium [2,6–8].

The Li-rich clay deposits of our study occur and are well-preserved in the Xiejiacao section of Guangan city, South China. Detailed investigations of those minerals are important for obtaining a better understanding of the formation mechanism of Li-rich K-bentonite deposits, particularly of the relationships between the source magmas, altered clay minerals, sedimentary environments, and diagenetic process in these clay deposits. The objectives of this study on Li-rich K-bentonite deposits of the Xiejiacao section are, therefore, to: (1) ascertain if the generic type of Li-rich deposits is volcanic origin or weathering origin; (2) define the clay mineralogy and associated non-clay minerals in detail; (2) characterize the distribution of major and trace elements in the Li-rich clay deposits; and (3) assess the source of the Li-rich K-bentonite deposits and their formation conditions.

## 2. Geological Setting

The Xiejiacao section is situated at Guangan City, Sichuan Province, South China, and tectonically located in the Sichuan Basin, Upper Yangtze Platform (UYP) (Figure 1). The Yangtze Plate and North China Plate collided diachronically from Late Permian to Triassic, resulting in the closure of the Mianxian–Lüeyang paleo-ocean [9,10]. The palaeogeographic pattern of the Yangtze Plate underwent significant changes in the Triassic. During the Early Triassic to Middle Triassic (Jialingjiang Formation, JF, T$_1$*j*), the sedimentary environment of the studied area became restricted due to the westward sea retreat caused by the uplift of the northern margin of the UYP [11]. The whole Sichuan Basin was uplifted due to Episode I of the Indosinian Orogeny by the end of the Middle Triassic, and the sea retreated from the southern part of the basin in this period, thereby exposing the Leikoupo Formation (LF, T$_2$*l*) to weathering and erosion for tens of thousands of years. Stratigraphically, the LF conformably overlies the Lower Triassic JF and sits beneath and parallel to the Upper Triassic Xujiahe Formation (XF, T$_3$*xj*) [12]. The JF of the UYP, corresponding to the Early Triassic Olenikian stage, is made up of a series of carbonates and evaporites up to 300–1000 m in thickness [13]. The stable UYP was mainly a cratonic carbonate platform during the deposition Middle Triassic LF (corresponding to the Middle Triassic Anisian Stage) and, therefore, limestone, dolomitic limestone, dolomite, gypsiferous dolomite, and gypsum widely formed in this stage [11].

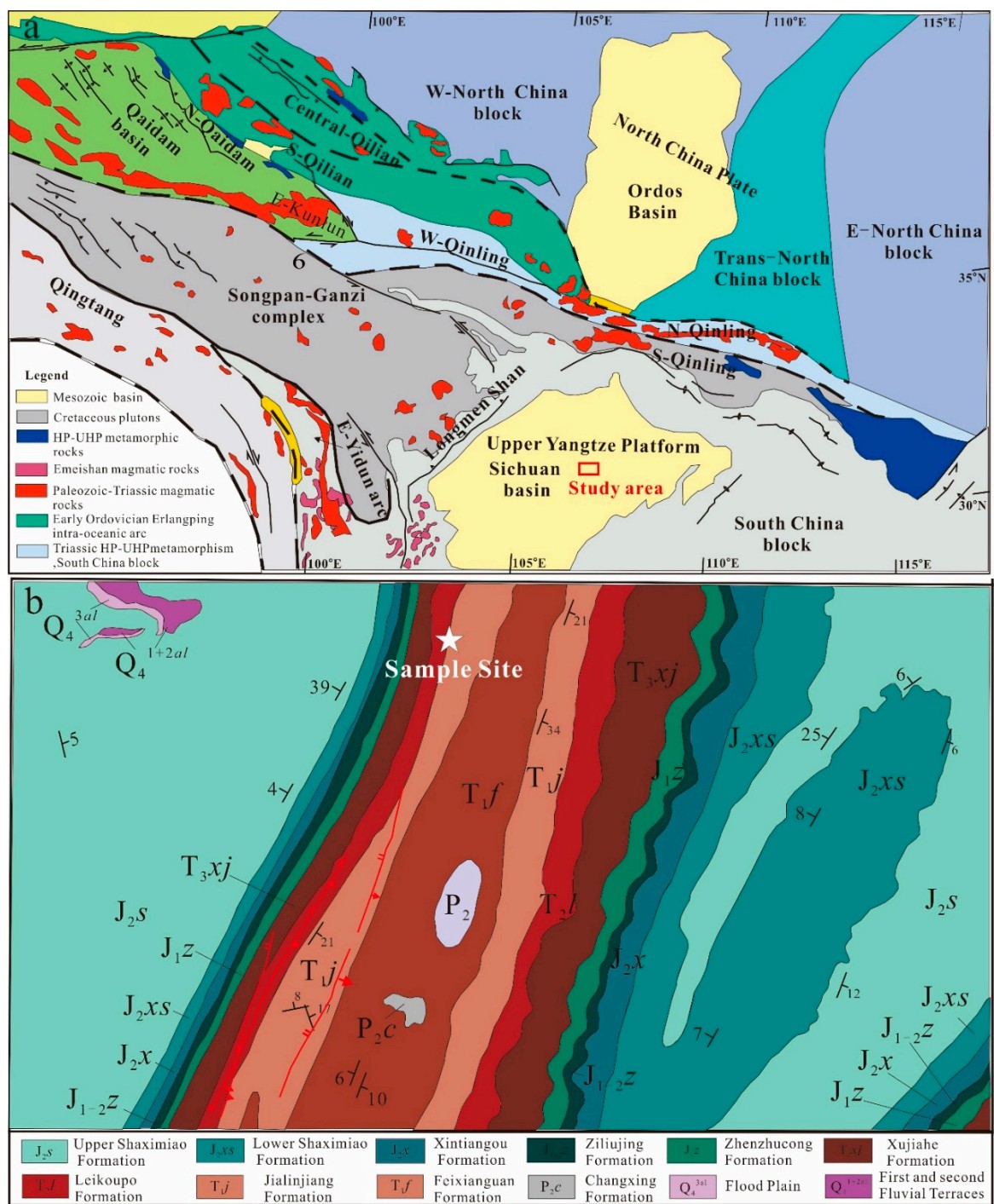

**Figure 1.** (**a**) Simplified regional map showing the locations and tectonics of the Sichuan Basin and its surrounding areas (modified from Enkelmann et al., 2007 [14]). (**b**) Simplified regional geological map showing the locations of Xiejiacao section in studied area [15].

The Li-rich K-bentonite deposits under study, which are composed of a lithology known as mung bean rock, are widely distributed in Early–Middle Triassic strata of the south China, over an area of ~700,000 km$^2$ and occurring in beds dozens of centimeters to tens of meters in thickness [4]. The Li-rich K-bentonite deposits of the Xiejiacao section is a cemented clay bed ~30 cm thick, exhibiting light green colors and hard characteristics, which has a typical texture of vitric tuff under the microscope (Figure 2). The Li-rich clay deposits overlies the JF and underlies the LF, and is generally considered to represent the boundary between Early and Middle Triassic. The limestones of the JF, below the bentonitic beds,

are rich in foraminifera indicating a late Olenekian stage [16,17]. However, Zircon U-Pb dating for one sample (XJC-1-R-1) by the laser ablation inductively coupled plasma mass spectrometry (LA-ICP-MS) method yielded one weighted mean $^{206}$Pb/$^{238}$U age of 225.9 ± 1.4 Ma (MSWD = 7.1, n = 29) [3], which is approximately Upper Triassic.

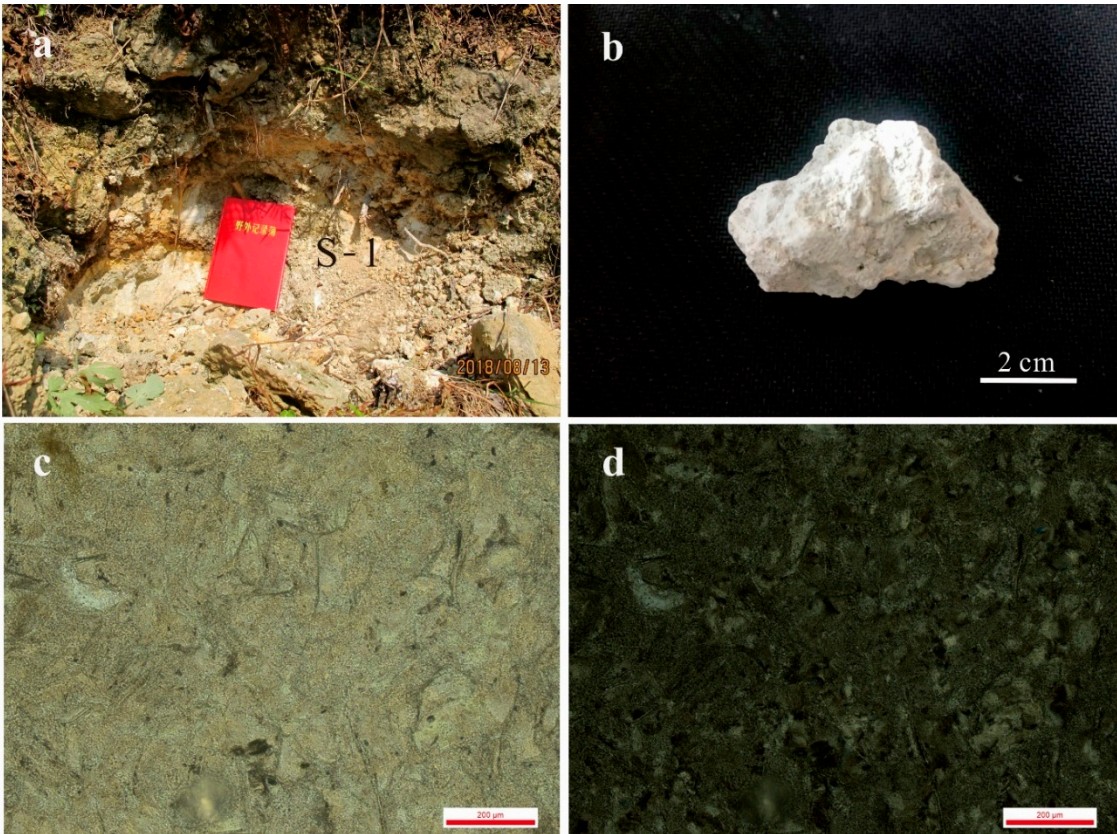

**Figure 2.** (**a**) Close-up photographs of the Li-rich K-bentonite deposits of Xiejiawan section; (**b**) a representative hand specimen; (**c**) microphotograph of Li-rich clays sample S-1 (in plane-polarized light); (**d**) microphotograph of Li-rich clays sample S-1 (in cross-polarized light).

## 3. Samples and Methods

The Li-rich K-bentonite deposits and JF limestone were collected from the Xiejiacao section, Guangan City, Sichuan Province, in May 2018. One mung bean rock sample (S-1) was collected after removal of the weathered surface crust, which is light green, consolidated and hard, containing elliptical siliceous particles locally (Figure 2). Three limestone samples were collected from the JF beneath the mung bean rock, and numbered S-2 to S-4. Sun et al. (2018) reported data for three further clay samples (TL-1~3) from the neighboring areas Tongliang section [2], while Ju et al. (2019) published data for one clay sample (XJC-1-R-1) from the same outcrop with this study in the Xiejiacao section [3]. The mineralogy of the whole rock and of the clay minerals on random specimens prepared by the side-loading method was carried out by X-ray diffraction (XRD) a using an X-ray diffractometer (TTR-3, Rigaku Corp, Tokyo, Japan). The XRD was performed at 45 kV and 30 mA with a Cu Kα radiation (λ = 1.54056 Å). The clay mineral fractions (<2 um) were extracted from the powder samples after centrifugation in distilled water according to the analytical procedure of Jackson et al. (1978) [18]. The XRD analyses were conducted on the air-dried oriented clay sample (AD), ethylene glycol-saturated clay sample (EG), and 550 °C heating clay sample, respectively. The mineral compositions and their relative proportions of the bulk rocks and clay minerals in the purified clay samples were obtained using the Clayquan (2016 version) program according to the XRD data of non-oriented powdered samples and oriented clay samples, respectively. The ratio of illite to smectite in the mixed layer

illite/smectite (I/S) was calculated by the following formula: I/(I/S) = $\frac{I_{10}(EG)}{I_{10}(550)-I_{10}(EG)}$. Here, I/(I/S) refers to the mineral contents ratio of illite to mixed layer I/S; $I_{10}(EG)$ indicates the peak area of the (001) peak in the ethylene glycol-saturated clay samples; $I_{10}(550)$ indicates the peak area of (001) for the sample after heating to 550 °C [19].

Chemical analyses of all bulk rocks were carried out using X-ray fluorescence spectrometry (XRF) for major elements and inductively coupled plasma mass spectrometry (ICP-MS) for trace elements. Pretreatment procedures were as described by Yang et al. (2007) [20]. Analytical precision was better than 5% for repeated analyses of Chinese national standards GB/T 14506.14-2010 and GB/T 14506.28-2010 [21,22].

Strontium was separated twice using a cation exchange procedure using 100-200 mesh AG58X8 resin [23] and measured using a Finnigan Triton Thermo ionization mass spectrometer (TIMS) at Nanjing University for isotopic research. The analytical procedure of strontium isotope were described by Hu et al. (2017) [24]. $^{87}Sr/^{86}Sr$ ratios were corrected for mass fractionation by normalizing to $^{86}Sr/^{88}Sr = 0.1194$ with exponential law. Repeated measurements of Sr standard NBS987 yielded $^{87}Sr/^{86}Sr = 0.710259 \pm 0.000015$ (2SD).

## 4. Results

### 4.1. Bulk and Clay Mineralogy

The XRD analyses of the bulk rock, AD-oriented clay sample, EG-saturated clay sample, and 550 °C heating clay sample are given in Figure 3. The bulk rock of the Li-rich clay sample(S-1) is composed of clay minerals (37.1%), quartz (33.9%), K-feldspar (19.4%), dolomite (6.6%) and calcite (3.0%). The clay minerals are composed mainly of illite (86%), with minor R3 ordering I/S (13%), and chlorite (1%) (Figure 3). The air-dried smectites usually show a peak at around 15 Å which changes to around 17 Å upon glycolation. However, the peak at 15–17 Å can only be observed in smectite and partially ordered interstratification I/S, and no peaks at 15–17 Å can be observed in ordered I/S [25]. Therefore, no discrete smectite was found to be present in the samples. The illite is non-expandable and characterized by an intense broad $d_{001}$ peak at around 10 Å, with further peaks at 5 Å and 3.33 Å in the low-angle region, which remain unchanged in all three XRD patterns. In general, the peaks of I/S, overlap with those of illite on air-dried oriented clay samples as well as the 550 °C heating clay sample. However, the I/S minerals of S-1 samples are characterized by a strong peak at around 10 Å in the air-dried clay fraction, which collapsed to 9.8966 Å and 10.160 Å after saturation with ethylene glycol and subsequently shrunk to 10 Å after heating at 550 °C for 1 h. Due to the presence of only one expandable mineral in the studied sample, the $2\theta$ technique can be reliably used to estimate %S (percent of smectite layers in I/S) [26]. The degree of long-range ordering represented by Reichweite (R) parameters varied consecutively from R0 to R3 via R1 and R2 with increasing illite content [27]. The lowest reliable %S of I/S in Li-rich clay deposits is around 15%, suggesting a high degree of I/S order (R3).

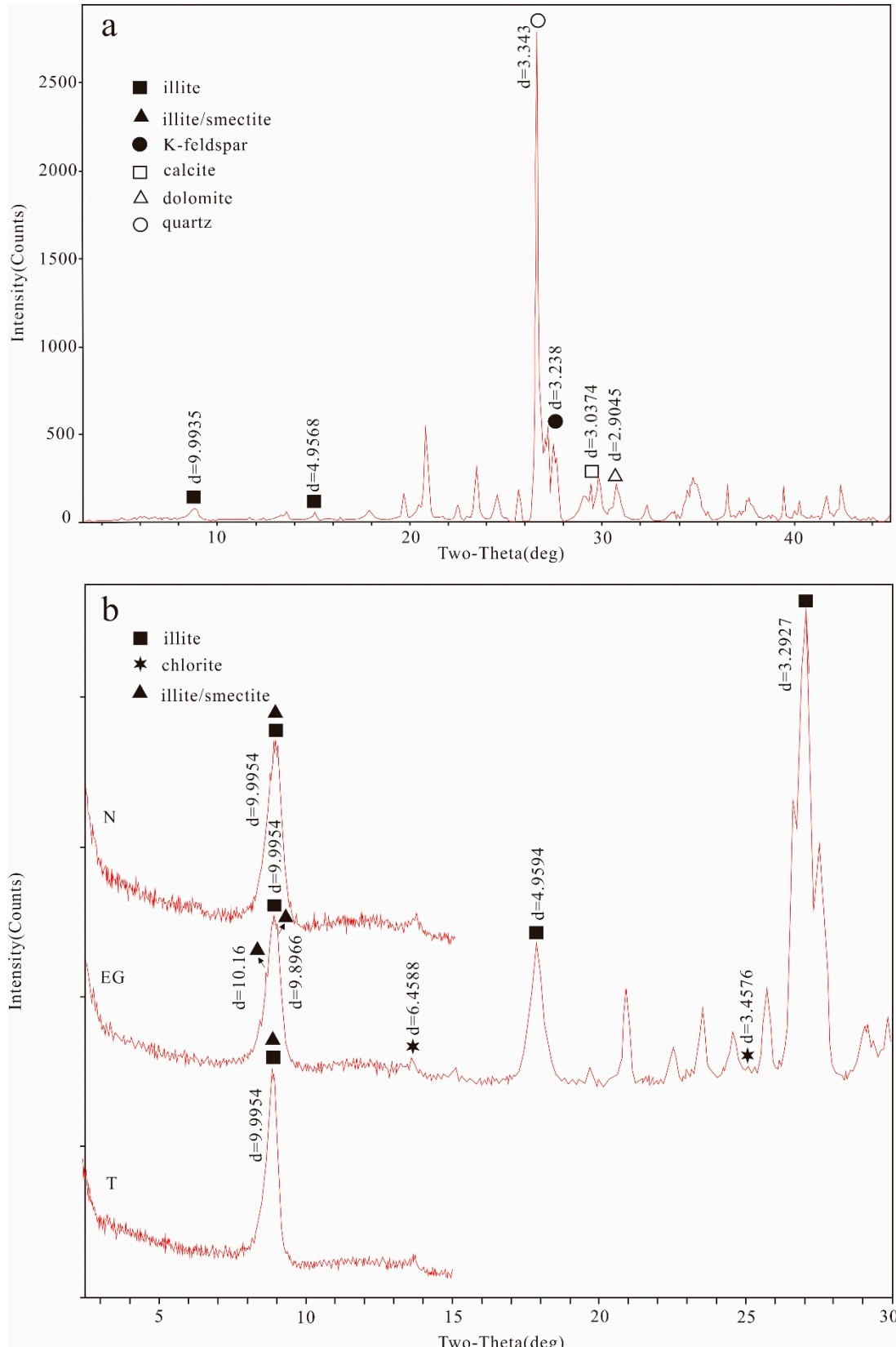

**Figure 3.** X-ray diffraction (XRD) patterns of the studied lithium-rich clay sample S-1 from the Xiejiacao section. (**a**) Bulk samples; (**b**) The diffraction pattern of clay minerals. N, Air-dried oriented clay samples; EG, Ethylene glycol-saturated clay samples; T, clay samples post heating to 550 °C.

*4.2. Elemental Geochemistry*

　　　The major and trace element compositions of the Li-rich clay sample (S-1) and limestones samples (S-2,3,4) in the studied area are presented in Tables 1 and 2 [3]. The most abundant major component of the limestone sample was CaO (45.68%~54.77%, average = 48.86%), followed by volatiles, measured as loss on ignition (LOI) (38.47~43.92%, average = 41.95%), $SiO_2$ (0.696%~9.93%, average = 4.075%), and MgO (0.373%~7.20%, average = 3.181%). The results for sample S-1 show that the contents of $SiO_2$, MgO (4.18%) and $K_2O$ (10.04%) are higher while the $Al_2O_3$ content (12.41%) is lower than the values reported for the post-Archean Australian shale (PAAS) and north American shale composite (NASC) [28]. The $K_2O$ contents of the ash samples are largely dependent on the presence of I/S and illite, since K is usually fixed in the interlayer sites of illite [29]. The illite typically contains 7% $K_2O$, and the $K_2O$ content of the ash sample will increase with increased proportion of illite layers in the mixed-layer I/S clays [30–32]. The $K_2O$ contents of the Li-rich clay sample in the Xiejiacao section are obviously higher than those of their limestone host rocks, indicating that K was probably sourced from the porewater and volcanic ashes during early diagenesis. Consequently, a large amount of porewater K is conducive to the formation of mixed-layer R3 ordered I/S clays with smectite layer contents of 15%.

　　　The primitive mantle (PM)–normalized and upper continental crust (UCC)–normalized trace element distributions of our samples are shown in Figure 4a,b [33]. The most abundant trace element of Li-rich K-bentonite deposits (S-1) is B (867 ppm), followed by Li (321ppm), Rb (122 ppm), Zr (102 ppm) and Ba (65.4 ppm). The trace elements in Li-rich K-bentonite deposits are characterized as being evidently depleted compared with the PM except for Li, B, Rb and Ba, and depleted compare with the UCC except for Li and B. The clays sample have an overall enrichment of Li and B of 100–1000 times compared to PM, and of 10–100 times compared to UCC. In addition, the clay samples are relatively weakly enriched in high field-strength elements (HFSE), and Zr, Ta, Nb and Hf show a slightly positive anomaly. The trace element compositions of limestone samples (S-3,4) are characterized as obviously depleted relative to the PM except for Li and B. The limestone sample S-2 has pronounced positive Sr anomalies with a highest Sr value of 1227 ppm.

　　　The rare earth element (REE) data of our sample in the studied area are listed in Table 3 and chondrite-normalized REE distribution patterns are presented in Figure 4c. The Eu and Ce anomalies were determined respectively by $Ce/Ce^* = (Ce_N)/[(La_N + Pr_N)^{1/2}]$ and $Eu/Eu^* = (Eu_N)/[(Sm_N + Tb_N)^{1/2}]$, in which the subscript N denotes normalization of the REE to chondrite [28]. The total REE content of clays sample (S-1) is 73.779 ppm, while limestone samples range from 3.254 to 18.676 ppm with an average of 9.041 ppm. The chondrite-normalize REE distribution patterns of clays and limestone samples show fractionated patterns with a negative Eu anomaly, while the clays sample (S-1) has much larger negative Eu anomalies than the limestone samples. The clay samples have negatively sloping curves with an overall enrichment of light rare earth elements (LREEs) of 10–100 times chondritic, and of heavy rare earth elements (HREEs) a factor of 10. In contrast, the limestone samples have an overall enrichment of LREEs of 1–10 times chondritic, but a deletion of HREEs a factor of 0.01-1. The average $(La_N/Yb_N)$ ratios of limestones and clays samples are 11.1289 and 3.7126 respectively, exhibiting a high degree of fraction between LREEs and HREEs.

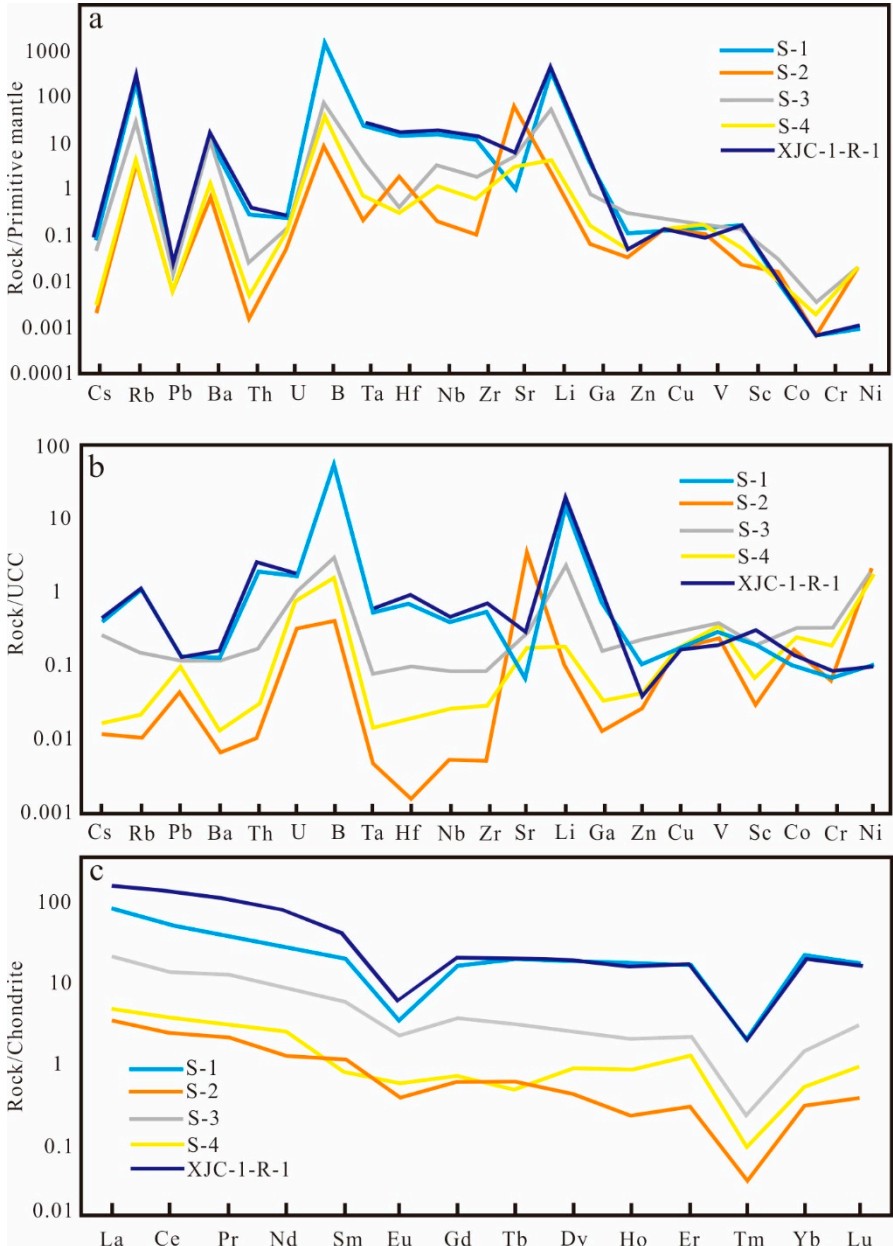

**Figure 4.** (**a**) Primitive-mantle (PM) normalized, and (**b**) upper continental crust (UCC) normalized trace element distributions, and (**c**) chondrite-normalized REE distributions of the lithium-rich K-bentonite deposits in Xiejiacao section, South China; Chondrite, PM and UCC normalizing values from Sum and McDonough [33].

**Table 1.** Major chemical compositions of the collected samples (wt %). Data for sample XJC-1-R-1 are from Ju et al., 2019 [3]; data for TL-1-3 are from Sun et al., 2018 [2].

| NO. | $SiO_2$ | $Al_2O_3$ | $Fe_2O_3$ | MgO | CaO | $Na_2O$ | $K_2O$ | MnO | $TiO_2$ | $P_2O_5$ | LOI | FeO | F |
|-----|------|-------|-------|------|-------|--------|-------|-------|-------|-------|-------|--------|-------|
| S-1 | 67.73 | 12.41 | 0.944 | 4.18 | 0.418 | 0.015 | 10.04 | 0.009 | 0.238 | 0.029 | 3.74 | 0.21 | 0.261 |
| S-2 | 0.696 | 0.154 | 0.075 | 0.373 | 54.77 | 0.089 | 0.082 | 0.005 | 0.006 | 0.014 | 43.46 | <0.10 | 0.044 |
| S-3 | 9.93 | 1.82 | 1.06 | 1.97 | 45.68 | 0.023 | 0.824 | 0.041 | 0.109 | 0.043 | 38.47 | 0.19 | 0.09 |
| S-4 | 1.6 | 0.402 | 0.357 | 7.2 | 46.13 | <0.010 | 0.281 | 0.015 | 0.032 | 0.031 | 43.92 | 0.19 | 0.045 |
| XJC-1-R-1 | 66.96 | 14.18 | 0.60 | 3.84 | 0.37 | 0.06 | 11.04 | 0.01 | 0.23 | 0.05 | 3.36 | - | - |
| TL-1 | 58.52 | 16.00 | 0.66 | 8.11 | 0.42 | <0.01 | 9.02 | <0.01 | 0.24 | 0.12 | - | - | - |
| TL-2 | 62.02 | 15.00 | 2.96 | 3.35 | 1.36 | <0.01 | 8.44 | 0.03 | 0.84 | 0.38 | - | - | - |
| TL-3 | 58.24 | 16.74 | 0.80 | 7.29 | 0.69 | <0.01 | 8.84 | <0.01 | 0.27 | 0.33 | - | - | - |

Note: All Fe as $Fe_2O_3$; LOI is loss on ignition.

**Table 2.** Concentrations of trace elements and REE of the collected samples (ppm). Data sources are described in the caption for Table 1.

| NO. | Cs | Rb | Pb | Ba | Th | U | B | Ta | Hf | Nb | Zr | Sr | Li | Ga | Zn | Cu | V | Sc | Co | Cr | Ni |
|-----|-----|-----|-----|-----|-----|-----|-----|-----|-----|-----|-----|-----|-----|-----|-----|-----|-----|-----|-----|-----|-----|
| S-1 | 1.52 | 122 | 2.69 | 65.4 | 19.8 | 4.47 | 867 | 1.14 | 4.1 | 9.63 | 102 | 22.4 | 321 | 13 | 7.08 | 4.1 | 17.3 | 2.12 | 0.97 | 2.37 | 2.02 |
| S-2 | 0.04 | 1.12 | 0.89 | 3.31 | 0.12 | 0.9 | 6 | 0.01 | 0.54 | 0.13 | 0.9 | 1227 | 1.96 | 0.22 | 1.75 | 4.36 | 15.5 | 0.31 | 1.67 | 2.18 | 39.7 |
| S-3 | 0.91 | 16.3 | 2.23 | 62 | 1.8 | 2.91 | 47.2 | 0.16 | 0.11 | 2.09 | 16.3 | 92.2 | 50.1 | 2.67 | 17 | 7.09 | 23.1 | 1.99 | 3.17 | 11.1 | 35.9 |
| S-4 | 0.06 | 2.34 | 1.85 | 6.83 | 0.33 | 2.23 | 25.5 | 0.03 | 0.08 | 0.68 | 5.2 | 60.7 | 3.68 | 0.53 | 2.94 | 4.4 | 21 | 0.72 | 2.35 | 6.7 | 35.7 |
| XJC-1-R-1 | 1.96 | 140.52 | 2.86 | 95.04 | 31 | 5.74 | - | 1.33 | 5.43 | 11.94 | 137.67 | 101.14 | 380.02 | 17.38 | 2.64 | 4.09 | 11.16 | 3.37 | 1.38 | 3.01 | 1.87 |
| TL-1 | 11.8 | 291 | 5.28 | 120 | 29.5 | 8.02 | - | 1.55 | 5.39 | 12.5 | 140 | 26.5 | 663 | 22.1 | 49.1 | 4.23 | 4.59 | 4.03 | 0.12 | - | - |
| TL-2 | 12.5 | 157 | 34.5 | 287 | 21.5 | 9.81 | - | 1.5 | 6.03 | 17.8 | 205 | 50.4 | 257 | 0.81 | 123 | 35.9 | 95.3 | 10.7 | 8.86 | - | - |
| TL-3 | 16.8 | 301 | 14.2 | 55.1 | 16.8 | 13.3 | - | 1.59 | 5.60 | 13.3 | 142 | 21.4 | 594 | 0.49 | 60.7 | 5.09 | 4.94 | 4.4 | 1.94 | - | - |

**Table 3.** Concentrations of rare earth elements of the collected samples (ppm). Data sources are described in the caption for Table 1.

| NO. | La | Ce | Pr | Nd | Sm | Eu | Gd | Tb | Dy | Y | Ho | Er | Tm | Yb | Lu |
|-----|-----|-----|-----|-----|-----|-----|-----|-----|-----|-----|-----|-----|-----|-----|-----|
| S-1 | 16.3 | 27.3 | 2.99 | 10.3 | 2.6 | 0.176 | 2.75 | 0.599 | 3.83 | 22.3 | 0.807 | 2.32 | 0.461 | 2.96 | 0.386 |
| S-2 | 4.36 | 6.95 | 1.01 | 3.45 | 0.762 | 0.114 | 0.641 | 0.098 | 0.553 | 2.8 | 0.098 | 0.309 | 0.054 | 0.21 | 0.067 |
| S-3 | 0.729 | 1.3 | 0.176 | 0.527 | 0.155 | 0.02 | 0.11 | 0.02 | 0.097 | 0.476 | 0.012 | 0.044 | 0.009 | 0.046 | 0.009 |
| S-4 | 1.02 | 2.06 | 0.26 | 1.01 | 0.11 | 0.03 | 0.131 | 0.016 | 0.208 | 0.998 | 0.042 | 0.182 | 0.023 | 0.079 | 0.022 |
| XJC-1-R-1 | 45.599 | 99.297 | 11.286 | 40.372 | 6.312 | 0.28 | 3.752 | 0.696 | 4.506 | 23.174 | 0.819 | 2.555 | 0.373 | 2.349 | 0.348 |
| TL-1 | 15.9 | 37.3 | 4.64 | 18.5 | 4.38 | 0.24 | 4.61 | 0.89 | 6.34 | 31.2 | 1.29 | 3.92 | 0.62 | 3.99 | 0.55 |
| TL-2 | 79.3 | 172 | 20.80 | 79.3 | 16.6 | 1.3 | 15.4 | 2.5 | 15.5 | 76.8 | 2.95 | 8.34 | 1.23 | 7.89 | 1.08 |
| TL-3 | 26.9 | 63.6 | 8.13 | 33.7 | 8.98 | 0.48 | 10.2 | 1.86 | 12.2 | 61.6 | 2.39 | 6.82 | 1.03 | 6.38 | 0.9 |

*4.3. Sr Isotopic Compositions*

The Li-rich K-bentonite sample has a notably higher $^{87}Sr/^{86}Sr$ value of 0.759758 compared to those of 0.708218 to 0.710737 for the limestone samples (Table 4). During the deposition of the JF (Lower Triassic), the $^{87}Sr/^{86}Sr$ ratios of coeval seawater have been previously reported to reach 0.70784 [34], which is approximately consistent with our data. The $^{87}Sr/^{86}Sr$ value of clays sample yield a positive anomaly, which is significantly higher than the amount that could be supplied by river (0.7116), benthic (0.7084) or hydrothermal (0.7037) sources [35]. However, this positive $^{87}Sr/^{86}Sr$ value is close to those of boundary sediments of volcanic origin [36,37].

**Table 4.** Sr isotopic compositions of the collected samples.

| Sample No. | Sample Type | $^{87}Sr/^{86}Sr$ | Std. Error |
|---|---|---|---|
| S-1 | Li-rich clays | 0.75975786 | 0.00000577 |
| S-2 | limestones | 0.70821791 | 0.00000562 |
| S-3 | limestones | 0.71073714 | 0.00000575 |
| S-4 | limestones | 0.70876008 | 0.00000603 |

## 5. Discussions

*5.1. Source Magmas and Tectonic Settings*

The major and trace element compositions of the clay sample (XJC-1-R-1) collected from the same outcrop of the Xiejiacao section and the clay sample (TL-1~3) collected from the Tongliang section in the neighboring area are also presented in Tables 1 and 2 [2,3]. The REE data of the clay sample (XJC-1-R-1) and (TL-1~3) are also listed in Table 3 and chondrite-normalized REE distribution patterns are presented in Figure 4c. Overall, the major element composition of the bulk rock (S-1) is consistent with the data for the clay sample (XJC-1-R-1) published by Ju et al. (2019) [3], which was collected from the same outcrop in the Xiejiacao section. HFSE, (e.g., Nb, Ti, Zr, Y, Ta) and REEs are widely considered to preserve source characteristics as they usually remain immobile during secondary alteration processes [38,39]. Therefore, the HFSE contents of bulk rocks are widely used to reveal the parental magmas and tectonic settings of volcanic ashes. The HFSE (Nb, Ta, Hf, Zr), REEs, and $TiO_2$ are considered important indicators for magmatic origin as they are immobile during diagenesis and weathering [40,41]. The $Al_2O_3/TiO_2$ ratio is commonly used as reliable proxy indictor of the provenance, since the contents of Al and Ti remain constant in materials with different degrees of weathering [42,43]. According to the classification model ($Al_2O_3$ vs $TiO_2$), the volcanic ashes corresponding to the Li-rich K-bentonite deposits of the Xiejiacao section (S-1 and XJC-1-R-1) and Tongliang section (TL-1~3) are classified as felsic magmas, with intermediate-acidic to acidic composition (Figure 5a). The $Zr/TiO_2$ and Nb/Y ratios are a reliable indicator of alkalinity and differentiation [38,44]. The Li-rich clay samples of Xiejiacao section and Tongliang section plot in the fields of rhyodacitic/dacitic based on the Winchester and Floyd (1977) [44] classification model, suggesting that the Li-rich clays probably originate from rhyodacite/dacite (Figure 5b). Following the discrimination diagrams after Pearce et al. (1984) [45], the immobile trace element Rb versus Y + Ta show that the Li-rich clays of Xiejiacao section and the Tongliang section lie in the field of within-plate granite, indicating that the parent volcanic eruption were caused by the mid plate volcanoes in the oceanic plate or associated with a mantle plume hotspot (Figure 5c). Furthermore, magmas that formed by a large fraction of low-temperature melts of felsic continental crust tend to have lower Zr concentration [1,46]. In addition, magmas with moderate to extreme lithium enrichment are generally considered to have incorporated felsic continental crust [1].

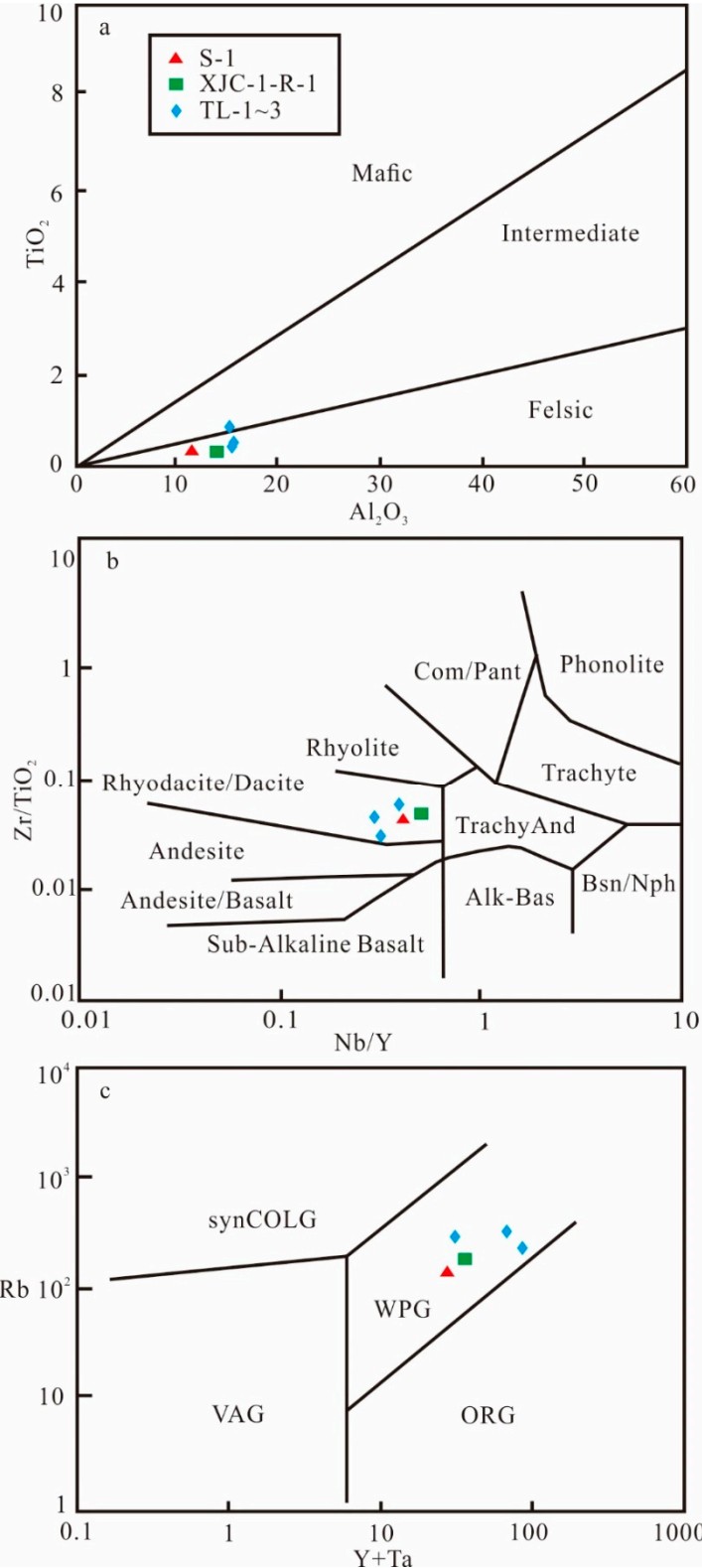

**Figure 5.** Li-rich K-bentonite deposits sample plot comparing data from Sun et al. 2018 [2], Ju et al. 2019 [3] and our data: (**a**) Plot of TiO$_2$ and Al$_2$O$_3$ (**b**) Bulk rocks ratios of Nb/Y and Zr/TiO$_2$ for Li-rich K-bentonite deposits from dashed lines indicate source fields defined by Winchester and Floyd (1977) [44]. (**c**) Rb versus Y + Ta plots of Li-rich K-bentonite deposits after Pearce et al. (1984) [45]. VAG = Volcanic arc granite; synCOLG = Syn-collision granite; WPG = within-plate granite; ORG = ocean ridge granite.

The Sr isotope signatures of samples from marine sections mainly depend on the Sr isotopic composition of seawater and the parent rock of the tuffs [38]. The Sr isotope composition of Li-rich K-bentonite deposits in the Xiejiacao section is 0.759758 ± 0.000012. The Sr isotope composition of underlying limestone samples (S-2~3) of JF range from 0.708218 to 0.710737, which represented the Sr isotopic composition of seawater at that time, far lower than that of the Li-rich clays. Thus, the high $^{87}Sr/^{86}Sr$ values of the Li -rich clays in this study can be attributed to the source parent rock. The increase in the $^{87}Sr/^{86}Sr$ value of clay deposits is mostly related to ignimbrite flare-ups with multiple caldera-forming eruptions, which exhibit a notably high $^{87}Sr/^{86}Sr$ value, corresponding to an increased input from a crustal source and decreased contribution of a mantle source [47]. In general, the rhyolite magmas derived from small proportions (~20%) of felsic continental crust have a high lithium concentration [1]. Therefore, partial melts in the continental crust were initially high in Li, B and other rare metals, which had preconcentrated in the deep stage magma before the volcanic eruption. Overall, the geochemical evidence suggests that the magmatic sources of Li-rich clays are mainly felsic, which most likely derived from a subduction-related volcanism [48,49].

*5.2. Formation Mechanism of Li-Rich Clay*

Fresh volcanic ash does not usually contain a significant amount of smectite. In deep-marine environments, volcanic ashes are precursor materials of bentonites, and thus bentonites have a diagenetic origin rather than a sedimentary origin. In this environment, the K, Ca and Mg are readily available and used for the formation of smectite and illite, thus the ash beds are mainly composed of smectite in the early stage and subsequently dominated by illite due to the illitization of smectite [50,51]. Although the mineralogy and chemistry of the clay minerals that form could be influenced by the variations of burial conditions, the volcanic ash is usually altered into smectite in subaqueous environments [49,52,53]. In this study, the clay minerals mainly consist of illite (86%), with minor R3 ordering illite/smectite (13%), and chlorite (1%). The clay minerals of Li-rich clays sample TL-1 and TL-2 from Tongliang section completely comprise I/S, while the sample TL-3 is mainly composed of R3 ordered I/S (68%), illite (24%) and chlorite (5%) [2]. Generally, the I/S in K-bentonite is considered to alter from the smectite during diagenesis, whereas smectite is mainly formed by submarine alteration of volcanic materials [52,54]. The illitization of smectite begins at about 70~80 °C and leads to a decrease in the percentage of smectite in this process, which is usually characterized by the following reaction sequence: smectite→randomly interstratified illite-smectite mixed layer (I-S)→ordered I-S→illite [55,56]. Although the composition and permeability of the parent ash seems to be the principal controlling factor in the illitization rates of smectite, it appears that other important factors could have influenced the reaction, such as the burial temperature and time, as well as the chemistry of porewaters (especially $K^+$ content) [56,57]. The characteristics of mixed layer I/S clay minerals are significantly affected by the deposition facies and porewater K concentrations [58]. The mixed-layer R3 ordered I/S derived from the smectite illitization of a volcanic ash bed observed in this study is probably attributed to deep burial and elevated alteration temperatures [59,60], while the poorly ordered (R0) mixed-layered I/S is usually attributed to a low burial temperature [61]. The temperatures and depth of post-sedimentary alteration reached by the sediments during the burial diagenesis history could be reliably estimated by the proportion of smectite layers in I/S [58]. The percent of smectite in I/S of the Li-rich K-bentonite is around 15%, suggesting that the temperatures of thermal evolution reached by the sediments are approximately 180 °C [55,62]. Based on the assumption of an average geothermal gradient of 30 °C/km in the Sichuan basin, the burial depth calculated for the Early–Middle Triassic Li-rich K-bentonites corresponds to around 6 km. The Eu/Eu* values of eruptive volcanic ashes are initially dependent on the growth of magmatic plagioclase, and the Eu generally tends not to fractionate relative to other REEs during the transport and deposition, except that it can easily be changed in a reducing diagenetic environment. The Eu anomaly, therefore, may be used as a robust proxy to indicate source-magma chemistry [63–65]. Felsic volcanic ashes usually yield a marked significant negative Eu anomaly, and the Li-rich K-bentonite (S-1) shows a negative Eu

anomaly (Eu/Eu* = 0.47), lower than that of typical felsic igneous rocks (0.86) [40,66,67], which suggests that the Eu is strongly fractionated in a deep burial reducing diagenetic conditions and thus $Eu^{3+}$ becomes more soluble $Eu^{2+}$. The absence of negative Ce anomalies in the Li-rich clays also indicates that the alteration took place in a suboxic or anoxic environment. The MgO content of Li-rich clays in this study is 4.18%, which is higher than the source magma of felsic volcanic rocks (0.4 to 1.8%) [68]. The Mg fixed in the clay minerals of felsic volcanic ash layers is generally considered to come mainly from the circulation between porewater (e.g., seawater) and the precursor material of Li-rich K-bentonite deposits during the post-sedimentary alteration [69–71]. The ash beds in this study are suggested to have formed in a marine sedimentary setting with a markedly high Mg content. In general, the smectite to chlorite transition is considered to mainly occur in three distinct geological environments: subaerial and submarine hydrothermal systems, sedimentary basins, and regional metamorphic terranes [72]. Smectite and chlorite are ubiquitous products of the diagenesis and low-temperature metamorphism of intermediate to mafic volcanic rocks and volcanogenic sediments [73]. In this study, large amounts of porewater Mg and Fe in alkaline marine environments probably drove the transition of smectite to chlorite [72,74].

## 6. Conclusions

The mineralogical and geochemical characteristics of Li-rich clay deposits from the Early–Middle Triassic portion of the Xiejiacao section, south China were investigated in detail in this work. Although the volcanic ashes examined have suffered significant diagenetic changes, geochemical and mineralogical analyses suggest that primary magmatic composition of this bed was mainly rhyodacitic/dacitic; trace-element data indicate the parent eruptive centers are characterized by the tectonic settings of a within-plate granite. Moreover, the precise determination of the I/S ratio reveals the thermal evolution of the sedimentary sequences reached approximately 180 °C corresponding to a burial depth of around 6 km. Overall, the mineralogical composition of the clay fraction indicates this lithium-rich clay deposit derived from volcanic materials shortly after deposition in deep marine conditions, and the clay fraction mainly contains illite and I/S clays derived from the volcanogenic smectite illitization. The Li in the interlayer space of illite and I/S is probably leached from the eruptive products by a mixed fluid source (i.e. meteoric, porewater and hydrothermal fluids) to become concentrated in the final clay minerals after post-sedimentary alteration.

**Author Contributions:** Conceptualization, M.Z., Y.Z. and Y.L.; investigation, J.Z. and J.X.; writing—original draft preparation, Y.L.; writing—review and editing, S.A.T.R.; supervision, M.Z.; project administration, E.X., X.N.; funding acquisition, Y.Z. All authors have read and agreed to the published version of manuscript.

**Funding:** This research was funded by subject of The National Key Research and Development Program of China (Grant Number: 2017YFC0602806), Projects of China Geological Survey (Grant Number: DD20190172) and Central Public-interest Scientific Institution Basal Research Fund (Grant Number: SYSCR2019-05).

**Acknowledgments:** We thank Lisa Stillings (U.S. Geological Survey), Linda Godfrey (Rutgers University) and the two anonymous reviewers for their insightful comments that greatly improved this manuscript.

**Conflicts of Interest:** The authors declare no conflict of interest.

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
