# Peer review of "Mineralogical and Geochemical Characteristics of Triassic Lithium-Rich K-Bentonite Deposits in Xiejiacao Section, South China"

_minerals, doi:10.3390/min10010069_

Round 1
Reviewer 1 Report
Paper: Mineralogical and Geochemical Characteristics of Triassic Lithium-rich K-bentonites Deposits in Xiejiacao Section, South China
Authors: Yongjie Lin et al.
The manuscript concerns mineralogical and geochemical investigations of Li-rich K-bentonites, South China. The subject od the manuscript interesting but this study contains very limited data. The main problem is insufficient sample numbers (only one sample from K-bentonites). Another problem is lacking the textural and geological data for the presence of K-bentonite. I have suspicious the studied sample (S-1) is a K-bentonite. The authors should prove that K-bentonite with textural data (thin-section photos). Photos from Figure 2 seem to be altered (pedogenic) clayey rocks. The determination of XRD patterns should be checked. there are many problems for mineralogical determinations as stated in review notes (see attached pdf file). In my opinion, this study needs more samples and careful determination of XRD patterns.
Finally, this manuscript is not suitable for Minerals Journal as present form. It has serious flaws and additional experiments needed, it needs more revision before resubmission.
Some specific comments:
Page 1, Line 23: Authigenic mineral should be occurred directly from diagenetic solutions (direct chemical precipitation) as a new components during diagenesis. Smectites should be occurred through the transformation from volcanic glass as related to volcanic ash sea water interaction.
Page 1, Line 28: K-bentonite term is not a mineral name. The authors should be given mineral name for Li fixation.
Samples and Methods section:
Page 3, Lines 83-84: The authors should prove that K-bentonite with textural data (thin-section photos). Photos from Figure 2 seem to be altered (pedogenic) clayey rocks.
Page 3, Line 89: What means "mung bean rock sample"?
Results Section:
Page 4, Lines 117-121: Figure and captions are completely wrong. The illustration of XRD pattern should be re-organized. Air-dried, glycolated and heated patterns should be shown in vertical order in order to understand the presence of I-S phase. In my opinion, the main clay phase is illite. I-S and chlorite are absent. K-feldspar term may be preferred instead of orthoclase. Orthoclase is characteristic for plutonic rocks, not for volcanic rocks. It should be sanidine.
Page 4, Lines 133-135:
(1) The opinion for source of K is speculative. K may release from K-feldspar and volcanic glass in tephra layer during diagenesis.
(2) The presence of R3 ordered I-S is speculative. There is no data in this study.
(3) The chemistry of clay fraction could not be evaluate for pure illite. Because clay fraction includes quartz and feldspar in addition to illite as seen in Fig. 2. Therefore this paragraph is not valid for illite chemistry.
Discussions Section:
Page 9, Line 232:
"The percent of smectite in I/S of the Li-rich K-bentonite is around 15%" How did you get this data? There is no data for this evaluation.

Author Response
Dear Professor,
Thank you for your comments concerning our manuscript entitled “Mineralogical and Geochemical Characteristics of Triassic Lithium-rich K-bentonites Deposits in Xiejiacao Section, South China” (minerals-582405). We thank for your great comments, which were helpful for further revising and improving our paper. We have made a revision according to all the comments as best as we can. Revised portions of the manuscript are highlighted in red and responses to the your comments are below.
Best regards,
Yongjie
Comments: Page 1, Line 23: Authigenic mineral should be occurred directly from diagenetic solutions (direct chemical precipitation) as a new components during diagenesis. Smectites should be occurred through the transformation from volcanic glass as related to volcanic ash sea water interaction.
Response: We have revised it, as follows:“The composition of the clay components suggests that the Li-rich clay deposits are probably altered from the smectite during diagenesis, whereas smectite is mainly formed by submarine alterations of volcanic materials and subsequently the I/S derived from the volcanogenic smectite illitization.”
Comments: Page 1, Line 28: K-bentonite term is not a mineral name. The authors should be given mineral name for Li fixation.
Response: We have revised it, as follows: “Lithium fixed in the illite and I/S is considered to have leached from the volcanogenic products by a mixed fluid source (i.e., meteoric, porewater and hydrothermal fluids).”
Samples and Methods section:
Comments: Page 3, Lines 83-84: The authors should prove that K-bentonite with textural data (thin-section photos). Photos from Figure 2 seem to be altered (pedogenic) clayey rocks.
Response: Thank you for this good comment. We have added two thin-section photos. This sample has a typical texture of vitric tuff under microscope, indicating an altered clay rock. According to the current situation, it is more appropriate to call it Li-rich clays than Li-rich K-bentonite.
Comments: Page 3, Line 89: What means "mung bean rock sample"?
Response: We described it in the introduction, as follows:““Mung bean rock”, a type of Li-rich clay deposits, is distributed widely in Early-Middle Triassic strata of south China, with a distribution area of around 700,000 km2,and dozens of centimeters to tens of meters in thickness[4]. It is so called “mung bean rock” because of its green color and often contains siliceous clasts[5,6] .”
Results Section:
Comments: Page 4, Lines 117-121: Figure and captions are completely wrong. The illustration of XRD pattern should be re-organized. Air-dried, glycolated and heated patterns should be shown in vertical order in order to understand the presence of I-S phase. In my opinion, the main clay phase is illite. I-S and chlorite are absent. K-feldspar term may be preferred instead of orthoclase. Orthoclase is characteristic for plutonic rocks, not for volcanic rocks. It should be sanidine.
Response: We have revised it.
Page 4, Lines 133-135:
Comments: (1) The opinion for source of K is speculative. K may release from K-feldspar and volcanic glass in tephra layer during diagenesis.
Response: We have revised it, as follows: “The K2O contents of Li-rich clay sample in the Xiejiacao section are obviously higher than those of their host rocks, indicating that K was probably removed from the porewater and volcanic ashes during early diagenesis.”
Comments: (2) The presence of R3 ordered I-S is speculative. There is no data in this study.
Response: We have added a description in 3. Samples and Methods and 4.1. Bulk and clay mineralogy section.
Comments: (3) The chemistry of clay fraction could not be evaluate for pure illite. Because clay fraction includes quartz and feldspar in addition to illite as seen in Fig. 2. Therefore this paragraph is not valid for illite chemistry.
Response: We have revised it.
Discussions Section:
Page 9, Line 232:
Comments:"The percent of smectite in I/S of the Li-rich K-bentonite is around 15%" How did you get this data? There is no data for this evaluation.
Response: The ratio of illite to smectite in the mixed layer I/S is calculated by the following formula: I/(I/S)=. In this formula, the I/(I/S) refers to the mineral contents ratio of illite to I/S; I10(EG) indicates the peak area of 10×10-1 on the ethylene glycol-saturated clay samples; I10(550) indicates the peak area of 10×10-1 on the 550℃ heating clay samples.

Reviewer 2 Report
The authors have presented a study of early mid-Triassic volcanic ash beds that occur in the Xiejiacao section, south China. Following multiple laboratory analyses their data show there are significant occurrences of lithium-rich K-bentonite deposits that may well have economic potential. The Li-rich K-bentonites contain quartz, clay minerals, feldspar, calcite and dolomite, and the clay minerals are dominated by illite and R3-ordered illite/smectite, characteristic of many well-documented K-bentonite deposits. The concentrations of major and trace element in Li-rich K-bentonite altered from volcanic ashes are most likely derived from felsic magmas, associated with intense volcanic arc activity. Using rare earth and high field strength element data the authors conclude that the Li-rich clays probably originated from rhyodacite/dacite source magmas. Based on widely accepted discrimination diagrams they indicate that the immobile trace element plots of Rb versus Y+Ta show that the Li-rich clays lie in the field of volcanic arc granites, indicating that the parent eruptive centers are characterized by the tectonic settings of a volcanic arc. This study is a very thorough investigation and will certainly be of interest to segments of the industrial community. However, the major problem that I find is the English grammar is in need of substantial correction. I have made a number of hand-written suggestions on the manuscript draft that I hope the authors find helpful.

Author Response
Dear Professor,
Thank you for your comments concerning our manuscript entitled “Mineralogical and Geochemical Characteristics of Triassic Lithium-rich K-bentonites Deposits in Xiejiacao Section, South China” (minerals-582405). We thank for your great comments, which were helpful for further revising and improving our paper. We have made a revision according to all the comments as best as we can. Revised portions of the manuscript are highlighted in red and responses to your comments as below.
Best regards,
Yongjie
1.Comments: However, the major problem that I find is the English grammar is in need of substantial correction. I have made a number of hand-written suggestions on the manuscript draft that I hope the authors find helpful.
Response: The entire manuscript has been revised to improve its readability.

Round 2
Reviewer 1 Report
The authors were revised the manuscript in terms of comments. Now it seems to be ready to accept for journal. According to my earlier comment (bentonite is not mineral name) the authors completely changed "K-bentonite term" with "clay deposit". This change is not necessary, K-bentonite term looks very appropriate instead of clay deposit because of indicating the origin of clay occurrences.
Before the acceptance of the manuscript, I would like to offer minor changes for two points:
(1) Please use "K-bentonite" term instead of "clay deposit" (except for mineral name).
(2) XRD data indicates the clay phase represented by completely illite mineral (I'm sure as a clya mineralogist). Please exclude or delete chlorite and illite-smectite (I-S) terms throught the figures and text.
Author Response
Dear Sir/Madam,
Thank you for your comments concerning our manuscript entitled “Mineralogical and Geochemical Characteristics of Triassic Lithium-rich K-bentonites Deposits in Xiejiacao Section, South China” (minerals-582405). We thank you for your great comments, which were helpful for further revising and improving our paper. We have made a revision according to all the comments as best as we can. Revised portions of the manuscript are highlighted in red and responses to your comments are below. The entire manuscript has been revised to improve its readability. We look forward to receiving your final decision regarding our manuscript.
Best regards,
Yongjie Lin
Responds to reviewer’s comments
Comments: Please use "K-bentonite" term instead of "clay deposit" (except for mineral name).
Response: Thank you for this good comment. We have changed “clay deposit” to “K-bentonite” in the manuscript, and the revised portions are highlighted in red
Comments: XRD data indicates the clay phase represented by completely illite mineral (I'm sure as a clya mineralogist). Please exclude or delete chlorite and illite-smectite (I-S) terms throught the figures and text.
Response: It is surely clear that reviewer has a strong background in clay mineral research. The Fig.3 with poor quality in manuscript seems to indicate the clay phase represented by illite mineral. We are very sorry for the poor-quality figures which bring the inconvenience to reviewer in the review process. According to your comments, we have checked the XRD data again in detail. We can be very sure that our results are correct, which are consistent with the data of clay sample (XJC-1-R-1) collected from the same outcrop of Xiejiacao section published by Ju et al. (2019). Both of these two studies confirm the presence of I/S in the clay sample. In this way, we have improved the quality of Fig.3 and attached the original XRD data for your check.

Reviewer 2 Report
The authors have presented a study of early mid-Triassic volcanic ash beds that occur in the Xiejiacao section, south China. Following multiple laboratory analyses their data show there are significant occurrences of lithium-rich K-bentonite deposits that may well have economic potential. The Li-rich K-bentonites contain quartz, clay minerals, feldspar, calcite and dolomite, and the clay minerals are dominated by illite and R3-ordered illite/smectite, characteristic of many well-documented K-bentonite deposits. The concentrations of major and trace element in Li-rich K-bentonite altered from volcanic ashes are most likely derived from felsic magmas, associated with intense volcanic arc activity. Using rare earth and high field strength element data the authors conclude that the Li-rich clays probably originated from rhyodacite/dacite source magmas. Based on widely accepted discrimination diagrams they indicate that the immobile trace element plots of Rb versus Y+Ta show that the Li-rich clays lie in the field of volcanic arc granites, indicating that the parent eruptive centers are characterized by the tectonic settings of a volcanic arc. Following a revision of the initial manuscript the authors have returned a revised version that has corrected all of the grammatical and syntactic errors. In my opinion the manuscript now reads very well and is suitable for publication.
Author Response
Dear Sir/Madam, September 31, 2019
Thank you for your comments concerning our manuscript entitled “Mineralogical and Geochemical Characteristics of Triassic Lithium-rich K-bentonites Deposits in Xiejiacao Section, South China” (minerals-582405). We thank you for your great comments, which were helpful for further revising and improving our paper. The entire manuscript has been revised to improve its readability.
Best regards,
Yongjie Lin

This manuscript is a resubmission of an earlier submission. The following is a list of the peer review reports and author responses from that submission.